# Examining Health Insurance and Non-Medical Challenges Among Vietnamese Americans in Texas During the COVID-19 Pandemic

**DOI:** 10.3390/ijerph22020189

**Published:** 2025-01-29

**Authors:** Alexander Le, Saba Siddiqi, Celine Nguyen, Ben King, Paul Gerardo Yeh, Jannette Diep, Lauren Gilbert, Bich-May Nguyen

**Affiliations:** 1Vietnamese Culture and Science Association, Houston, TX 77036, USA; alexle815@tamu.edu (A.L.); celine.nguyen@utsouthwestern.edu (C.N.); 2Texas A&M University College of Medicine, Bryan, TX 77807, USA; 3Tilman J. Fertitta Family College of Medicine, University of Houston, Houston, TX 77204, USA; ssiddi41@cougarnet.uh.edu (S.S.); kingb@central.uh.edu (B.K.); pyeh2@central.uh.edu (P.G.Y.); lrgilbe2@central.uh.edu (L.G.); 4University of Texas Southwestern Medical School, Dallas, TX 75390, USA; 5Boat People SOS Houston, Houston, TX 77072, USA; jannette.diep@bpsos.org

**Keywords:** health disparities, health insurance, minority health, public health, social determinants of health

## Abstract

When COVID-19 data on Asian Americans are available, they are frequently aggregated, concealing community-specific concerns. Consequently, there is limited COVID-19 literature on Vietnamese Americans. In this study, we investigated the association between health insurance coverage and non-medical challenges during the COVID-19 pandemic, in Vietnamese Americans in Texas. The NIH Community Engagement Alliance (CEAL) Common Survey 2 was administered electronically in English and Vietnamese and contained 23 questions about non-medical drivers of health, COVID-19 vaccination, and research participation. Vietnamese American adults in Texas were recruited between September 2021 and March 2022 via partnerships with community organizations. Responses were compared and analyzed using logistic regression. Of 217 respondents, 23 (11%) were uninsured. Of the uninsured participants, 43% lost health insurance coverage during the COVID-19 pandemic. Uninsured individuals had significantly higher odds of experiencing non-medical challenges, including obtaining housing (OR = 6.10, *p* < 0.001), food (OR = 6.41, *p* < 0.001), and medications (OR = 3.45, *p* < 0.05) than insured individuals. Uninsured individuals had a significantly longer time-lapse since seeing a healthcare provider (ordinal OR = 0.20, *p* < 0.05) than insured individuals. Thus, lack of insurance is strongly associated with non-medical challenges during the COVID-19 pandemic among Vietnamese Americans in Texas. Disaggregating data can address non-medical drivers of health, advancing equity for marginalized communities.

## 1. Introduction

The COVID-19 pandemic revealed and exacerbated the health disparities that exist among different racial and ethnic groups in the United States [1,2]. The pandemic disproportionately affected Asian Americans and Native Hawaiian/Pacific Islanders (AANHPI) both in terms of health outcomes and socio-economic impacts [2,3]. However, they are often overlooked or aggregated as a monolithic group in COVID-19 research, masking the diversity and heterogeneity within this population [4,5,6,7]. Consequently, the literature is scarce about the unique non-medical challenges during the COVID-19 pandemic that distinct AANHPI subgroups faced and continue to experience.

AANHPI are the fastest-growing racial or ethnic group in the United States (U.S.), having doubled from 10.5 million to nearly 20 million between 2000 and 2020 [8]. Using “Asian American” as an umbrella term to represent people who come from a diverse collection of sub-populations with distinct cultural and historical profiles can lead to stereotyping AANHPI as the “model minority” and obscures the disparities between AANHPI subgroups [9]. Disaggregating data is essential to identify the challenges faced by different communities.

Within this population, 2.3 million people are of Vietnamese descent, with nearly 300,000 living in Texas [10,11]. Vietnamese Americans, primarily arriving in the U.S. as refugees after the Vietnam War, have historically encountered challenges to healthcare access, such as language, education, poverty, and immigration status, contributing to wider health disparities [12,13,14,15,16,17]. These disparities are compounded by state-specific structural barriers within the U.S. healthcare system, with significant variation in coverage across the country. For example, in Texas, healthcare is largely privatized with limited expansion of Medicaid under the Affordable Care Act, and the state has the highest uninsured rate in the nation [18].

The COVID-19 pandemic, introducing unprecedented stress to healthcare systems globally, may have exacerbated these pre-existing barriers, especially among those who are uninsured or underinsured and those within immigrant and minority populations who may be obscured within monolithic, aggregated categories [19,20,21]. The COVID-19 pandemic created a global health crisis that impacted the world economically, psychologically, and socially. Economically, the pandemic led to a global recession with lockdown protocols, increased unemployment rates, and disruptions in product supply, leading to inflation in the cost of goods. The pandemic also had a social impact which can be highlighted by increases in family violence, insufficient remote education, and food insecurity within disadvantaged families from lack of school-supplied meals [22,23]. Marginalized communities were disproportionally impacted by the economic consequences of the pandemic. COVID-19-related challenges encompass the pre-existing inequities in social determinants of health that were intensified by the pandemic [24,25].

Health insurance is a key determinant of health, impacting access to healthcare services, preventive care, and treatment for COVID-19 and other chronic conditions [26,27]. Compared to other AANHPI subgroups who are more reliant on family-based immigration, Vietnamese Americans and other subgroups who primarily came as refugees are more likely to be insured but are less likely to have an employer-sponsored health plan [28,29]. However, limited literature exists on how insurance coverage has changed for Vietnamese Americans or Asian Americans during the pandemic and related impacts on obtaining essential resources. Some studies have found that Asian Americans, especially those who have completed less education, disproportionately experienced more pandemic-related job losses than other minority groups [3,30]. Further disaggregated research would provide more insights on Vietnamese American health insurance coverage amid COVID-19-related economic disruption and job loss, especially with Asian Americans being the most economically divided group in the United States [21,31].

Research examining the impacts of the COVID-19 pandemic on aggregated racial groups has revealed that the pandemic exacerbated existing challenges to insurance coverage that predated the COVID-19 era, with Hispanic/Latino and African American respondents more likely to attribute reduced healthcare access to the pandemic. Various pre-existing social determinants of health were affected by the COVID-19 pandemic, which had a universal impact, especially affecting marginalized communities. Along with Vietnamese Americans, the African American population faced concerns regarding mental health and wellness, limited healthcare accessibility and utilization, and disruption of social factors within the community [32].

Disaggregated data reveal that AANHPI subgroups faced significant health and social challenges, both during and prior to the pandemic [33,34]. For example, in limited research that reported disaggregated data of AANHPI populations, Vietnamese Americans were overrepresented in COVID-19 cases relative to other Asian subgroups [35]. These data parallel other disparities in AANHPI health outcomes. While pre-pandemic age-adjusted cancer mortality rates for the overall AANHPI population were lower than that of non-Hispanic whites, this aggregated statistic obscures that the incidences of certain cancers are higher for certain disaggregated AANHPI subgroups. Korean, Vietnamese, and Laotian Americans have also experienced increasing rates of breast and colorectal cancer—with both cancers being either stable or declining for the U.S. population [36,37]. On aggregate, Asian Americans of lower socioeconomic status are less likely to receive recommended preventative cancer screening, largely due to reduced access to health insurance [38].

For all racial groups, there is still a scarcity of literature that adequately disaggregates health and socioeconomic characteristics, including insurance, into ethnic subgroups, resulting in an incomplete picture of the pandemic’s impacts on insurance coverage and associated non-medical drivers of health for all populations [39,40].

The National Institutes of Health (NIH) Community Engagement Alliance (CEAL) collaborated with communities disproportionately affected by COVID-19 to promote testing, vaccinations, and clinical trial participation [41]. Through partnerships with the Texas CEAL Consortium and community-based organizations (CBOs), we investigated health insurance coverage and non-medical challenges, such as obtaining housing, food, and medications, during the COVID-19 pandemic among Vietnamese Americans in Texas. We examined the relationship between these challenges and health insurance status, as well as other demographic factors, including age, language proficiency, and education level.

Within this context, the study aims to explore how insurance access and non-medical drivers of health intersected to compound disparities during this period of heightened systemic stress.

## 2. Materials and Methods

The NIH CEAL collaborative developed the Common Survey 2 instrument, with Tier 1 comprising 23 questions about healthcare access, insurance status, risk perception, testing and disease control, and non-medical challenges [42].

### 2.1. Dependent Variables

Regarding healthcare, participants were asked how long it had been since they last saw a doctor or other healthcare professional about their health. Participants were asked to assess if they experienced non-medical challenges on a 3-point Likert Scale of “Yes, this is a major challenge” (3), “Yes, this is a minor challenge” (2), and “No, this is not a challenge” (1). Participants were also given the option of “Prefer not to answer”. Challenges assessed included “Getting the health care I need (including mental health)”, “Having a place to live”, “Getting enough food to eat”, “Having clean water to drink”, “Getting the medications I need”, “Getting where I need to go”, and “Taking care of my children or other people in my care”. Regarding vaccination status, participants were asked if they had received at least one dose of the COVID-19 vaccine previously.

Additionally, participants were asked to indicate whether they had health insurance. If yes, they were prompted to specify their primary type of health insurance or healthcare plan. Options included private health insurance through a job or school, insurance bought through a government exchange such as healthcare.gov, or insurance bought from a health plan or company, Medicare, Medi-Gap, Medicaid, CHIP, or kid’s state insurance, Military healthcare, and Indian Health Services. Participants were also provided with the option to select “Other” and manually input their health insurance, “Don’t know”, or “Prefer not to answer”. If none, loss of health coverage during the COVID-19 pandemic was assessed.

### 2.2. Independent Variables

Sociodemographic characteristics included gender (Female/Male/Other), sexual orientation (Heterosexual/Homosexual/Other), age groups (18–24, 25–34, 35–44, 45–54, 55–64, 65–74, 75+), preferred language for survey completion (English/Vietnamese), insurance status (Insured/Uninsured), and education (Less than a college degree/College degree or higher).

### 2.3. Study Eligibility, Procedures, and Recruitment

The translation, recruitment, consent, and sampling processes are described in more detail in a previous article [43]. CBO partners translated, back-translated, and reconciled translations of the instrument so that it was disseminated in English and Vietnamese. A convenience sampling approach was used to recruit Vietnamese American participants aged 18 and older residing in Texas who possessed proficiency in either English or Vietnamese reading and writing. The survey was open from 20 September 2021 to 4 March 2022. Recruitment efforts in English and Vietnamese occurred through email lists, social media, a webinar, and an Internet advertisement, as well as in-person at CBO literacy classes, health fairs, and clinics with large Vietnamese American populations.

As an incentive, participants who completed the survey and provided a valid email address or phone number were entered into a raffle for a chance to win one of five USD 50 credit card gift cards. This research received approval from the University of Houston Institutional Review Board (4 June 2021; STUDY00003046).

### 2.4. Statistical Analysis

We analyzed data using STATA (v17, College Station, TX, USA). Ordinal-level survey items, including those about non-medical challenges, were collapsed into dichotomous categories: Responses of “Yes, this is a major challenge” and “Yes, this is a minor challenge” were grouped as “agreement”, while “No, this is not a challenge” and “Prefer not to answer” were categorized as “neutral or disagreement”. All responses from the population of interest who completed their health insurance information were included using an available case analysis approach. All models were unadjusted bivariate tests of association; therefore, issues related to model fit, variance inflation, and parsimony of models were not a concern.

## 3. Results

In total, 217 adults responded to the health insurance questions in the survey. Table 1 provides the sociodemographic characteristics of the respondents, while Table 2 provides a logistic regression detailing the differences in non-medical challenges between insured and uninsured respondents.

### 3.1. Sociodemographic Characteristics

Table 1 provides sociodemographic information, including gender, sexual orientation, age, preferred language to use during the survey, insurance status, and education, from the online survey for Vietnamese adults living in Texas, which collected data from September 2021 to March 2022. The mean age of the sample was 47.6 years (median = 47, IQR: 33–63) and the average household size was four individuals. Overall, 25% of respondents preferred to use the surveys translated into Vietnamese language and about half had a college degree. Notably, nearly 96% of respondents had received at least one dose of the COVID-19 vaccine.

### 3.2. Non-Medical Challenges

Many participants reported experiencing non-medical challenges during the COVID-19 pandemic, including challenges in obtaining transportation (64.4%), healthcare including mental healthcare (57.3%), food (47.3%), medications (44.0%), clean water (39.8%), housing (37.0%), and care for dependents (40.8%). Figure 1 provides a forest plot comparing the differences in non-medical challenges between insured and uninsured individuals.

### 3.3. Insurance

Most participants were insured (87.3%). Of those who were uninsured (10.6%), a significant proportion had lost coverage during the COVID-19 pandemic (43%). Uninsured individuals had significantly higher odds of experiencing non-medical challenges, including obtaining housing (OR = 0.164, *p* < 0.001), food (OR = 0.156, *p* < 0.001), and medications (OR = 0.290, *p* < 0.007) when compared to insured individuals. Compared to those with insurance, individuals without insurance had significantly lower odds of getting vaccinated for COVID-19 (OR = 0.23, *p* < 0.05). When compared to insured individuals, those who were uninsured had a significantly longer time-lapse since they saw a healthcare provider (OR = 0.20, *p* < 0.05). Notably, there was no significant difference in age, preferred language, or educational attainment between insured and uninsured individuals.

## 4. Discussion

This study investigated the attitudes and perceptions among Vietnamese Americans regarding their health insurance coverage and non-medical challenges, such as obtaining housing, food, medications, and vaccinations, during the COVID-19 pandemic. At the time of the analysis, it was novel as the first exploration of this association, which remains understudied in the health disparities literature due to data aggregation under the AANHPI umbrella.

Approximately 10.6% of the survey respondents were uninsured. This percentage is higher than the overall percentage of uninsured Vietnamese Americans nationally, which was 8% in 2020 [28]. Uninsured adults are less likely than those with insurance to receive preventive and screening services, especially in a timely manner [42,44]. Health insurance plays a protective role in access to essential healthcare, as it can play a critical role in reducing out-of-pocket expenditures for health maintenance. One study found that families with an uninsured member had a higher unmet need for care due to costs than families where every member is insured [45]. The economic disruption of the COVID-19 pandemic resulted in millions of Americans losing their work or work-related income, further compounding healthcare affordability challenges. This loss disproportionately impacted people of color, including Asian Americans and individuals with lower incomes [46].

This study found that uninsured participants had significantly increased difficulties in obtaining medications, consistent with previous quantitative research on ethnic minorities. As of 2022, 53.4% of non-elderly Texans were estimated to have employer-based health coverage, either through their own job or as a dependent, while 18.9% were uninsured [47]. While Texas-specific data regarding Vietnamese American health coverage were relatively lacking, one report found that Vietnamese Americans in Texas have slightly higher uninsured rates when compared to the average total population and to Asian Americans on aggregate [48]. Similarly, national data suggest that Vietnamese Americans and other groups who largely arrived in the U.S. as refugees, such as the Bhutanese, Hmong, and Lao, are more likely to be covered by public insurance programs, such as Medicare and Medicaid, as compared to Asian groups that are more reliant on family-based immigration and, thus, more subject to restrictions regarding residency requirements [28,29]. As a result, these groups may have lower health insurance coverage rates and may rely more on private insurance or remain uninsured. Significantly longer lapses in visiting a healthcare provider for uninsured individuals are also in line with previous studies on this association [49].

Beyond the unmet need for health services, lack of health insurance is often additionally compounded by hardship regarding structural determinants of health. The participants without insurance reported increased hardship in securing housing, clean water, and food. The available literature on this association supports this finding. For example, Kuroki and Liu found a causal relationship between expanded health insurance coverage and increased homeownership rates among the low-income population [50]. Similarly, Himmelstein found that increased health insurance coverage via state Medicaid expansion led to reductions in food insecurity, likely due to reduced out-of-pocket healthcare expenses [51]. Water hardship, which encompasses incomplete plumbing and poor drinking water quality, has not been directly associated with health insurance in the literature but has been linked to inadequate housing and poverty, both factors correlating with health insurance coverage [52].

Barriers such as limited English proficiency and complex application processes may hinder access to enrollment in health insurance and federal assistance programs, particularly within linguistically heterogeneous AANHPI communities [53]. However, CBOs that have extensive reach among Vietnamese populations can build upon long-standing relationships to facilitate increased enrollment and participation. Many CBOs have had language-accessible programs and trained community navigators who have been historically associated with higher enrollment rates in public insurance programs among underserved and AANHPI populations [54,55,56]. For example, Vu et al. found that limited English proficiency was not a barrier to Supplemental Nutrition Assistance Program (SNAP) enrollment for Vietnamese Americans in California due to the efforts of both extensive outreach by CBO and inclusive state policies regarding language accessibility [57].

During the onset of the COVID-19 pandemic, CBOs across the country played leading roles in providing culturally adapted, in-language programming and information-sharing to AANHPI communities, including through the direct resource provision of food, free COVID-19 testing, and mobile vaccination sites [55]. Tailoring outreach efforts to incorporate community participation may be effective in empowering Vietnamese Americans and building resilience in responding to various challenges associated with a lack of insurance and other non-medical drivers of health, particularly during times of heightened systemic stressors.

Of our respondents, 96% had received at least one dose of the COVID-19 vaccine, despite concerns relating to healthcare access and insurance coverage. In researching Vietnamese Americans’ willingness to vaccinate, there are multiple factors that have been identified, including fear of severe illness, accessibility of obtaining the COVID vaccine within their communities, and trust in the research behind the vaccine, as well as in the medical professionals, faith-based organizations, and government entities that support it. In addition, obtaining the COVID-19 vaccine here in the United States was viewed as a privilege compared to other countries, such as Vietnam. These factors facilitated the decision of many Vietnamese Americans to outweigh the risks and barriers and ultimately receive the vaccine [58].

This high vaccination rate may also reflect the involvement of a community partner, Boat People SOS Houston (BPSOSH), which actively worked to connect this community with vaccinations during the pandemic, while simultaneously recruiting individuals for the survey. Through their social services and programming, BPSOSH was instrumental in directly working with community members in providing culturally and linguistically tailored information to encourage vaccine uptake. Moreover, the CBO noted that it was particularly challenging to recruit Vietnamese Americans who were unvaccinated for this study, as many unvaccinated individuals were reluctant to participate in the research. This observation aligns with anecdotal findings that rates of consent refusal for COVID–19–related randomized controlled trials have increased substantially during the pandemic [59]. Studies have linked such hesitancy regarding research recruitment to mistrust of institutions and perceived benefits of participation [43,60]. This difficulty underscores a potential selection bias in our sample, which makes it challenging to fully understand the barriers to healthcare and insurance coverage faced by unvaccinated Vietnamese Americans.

While this study focuses on minority health disparities and insurance access, we recognize that these issues exist within the larger context of pandemic-induced strain on healthcare systems. During the early phases of the COVID-19 pandemic, hospitals and healthcare systems globally faced significant challenges in balancing resources for COVID-19-related care and non-COVID-related healthcare services. Thus, these pre-existing structural inequities, such as limited insurance coverage, language barriers, and economic challenges, that disproportionately impact minority populations were likely exacerbated by the systemic difficulties facing healthcare systems during the pandemic [61].

Moreover, the impacts of the COVID-19 pandemic have highlighted persistent structural inequities faced by AANHPI populations, stemming from factors such as disproportionate representation in essential roles in the workforce, economic barriers, and multigenerational living arrangements—all of which contribute to AANPHI populations being at a higher risk of COVID-19 transmission [34]. However, studies have also found that these aggregated AANHPI data can obscure variations in infection, hospitalization, and mortality rates by subgroup [62]. For instance, among subcategories of Asian populations, Vietnamese and Filipino people were overrepresented in COVID-19 cases in Hawaii [35].

Disaggregated data regarding Vietnamese Americans would help unveil key disparities. In Santa Clara County, which has the largest Vietnamese American population in the United States, Vietnamese Americans had higher rates of COVID-19 cases than other Asian subgroups [63]. Other studies that used pre-pandemic disaggregated data have found that Vietnamese Americans have higher rates of cancer mortality, hypertension, and smoking prevalence among Asian subgroups [15,64]. In relation to non-medical challenges prior to the pandemic, limited studies suggest that Vietnamese Americans have lower rates of health literacy, higher rates of being housing cost-burdened (among mortgagors), and a higher percentage being low-income compared to both the general population and Asian Americans on aggregate [48,65]. To date, few large-scale studies have investigated chronic diseases or non-medical drivers of health within Vietnamese or other AANHPI subgroups using disaggregated data, likely due to limited availability.

Beyond AANHPI populations, data disaggregation can be a critical tool in understanding variations in outcomes within other racial and ethnic groups. However, disaggregated studies remain scarce. One limited study found that Dominican immigrants followed COVID-19 pandemic mitigation protocols in higher proportions than U.S.-born Dominicans [66]. Another study investigating acceptance of the COVID-19 vaccine found differing predominant motivations between Black ethnic subgroups—namely, African Americans, Caribbean Americans, and Black Africans [67].

Additionally, health insurance coverage for Latino subgroups varies widely, with Puerto Ricans having a low uninsured rate largely due to Puerto Rico’s status as a U.S. territory and its residents with citizenship avoiding immigration-related challenges to coverage. However, residents from certain Central American countries, including the “Northern Triangle” of Honduras, El Salvador, and Guatemala, are more likely to be undocumented and consequently face higher barriers to health insurance eligibility [40,68]. These findings underscore the importance of disaggregated data to reveal the unique challenges and disparities faced by specific subgroups within broader racial and ethnic categories.

### Strengths and Limitations

This study contains some limitations and challenges. The survey instrument addresses information related to COVID-19, limiting its generalizability to other conditions, and the study results do not establish causality. Additionally, this survey instrument does not ask participants about when their non-medical challenges began (i.e., before or during the COVID-19 pandemic) or if they directly attribute their associated difficulties to the onset of the pandemic. As a result, the results are unable to establish whether participants’ reported difficulties were pre-existing.

Furthermore, the sample is limited to self-identified Vietnamese Americans in Texas who were able to complete an online survey, introducing biases favoring technologically proficient, younger, and more acculturated individuals with access to technology. Moreover, this study overrepresents highly educated and English-proficient respondents, which may introduce bias by leaving out individuals with lower education levels or diverse backgrounds. Participants choosing “prefer not to answer” or “no opinion” were removed from the analysis, potentially introducing bias by not fully representing diverse opinions within the population of interest. These factors contributed to this small sample size which further limits the generalizability of the results. Compared to the population size of Vietnamese Americans residing in Texas, the sample size of 217 is limited.

Due to the recruitment of participants from health fairs, clinics, literacy classes, email lists, social media, webinars, and internet advertisements, we have an overrepresentation of a limited population margin. Additionally, the involvement of CBOs in both recruitment and connecting the community with vaccinations may have also contributed to potential bias, as the CBO played a crucial role in facilitating vaccine uptake among participants. Furthermore, unvaccinated individuals were often less willing to participate in COVID-19-related research, which led to a possible selection bias, as those unvaccinated were underrepresented in this study. These factors may explain why 96% of the sample population reported receiving at least one COVID vaccination.

Given the relatively small number of uninsured respondents (*n* = 23), subgroup analyses were limited, and the findings may not be generalizable to all uninsured Vietnamese Americans. Future research with larger samples will be needed to explore the association of health insurance coverage with non-medical challenges across different subgroups in Vietnamese American populations. This study addresses a significant gap in the health disparities literature by exploring the association between the COVID-19 pandemic and health insurance coverage, specifically within the Vietnamese American population. As communities move forward, lack of insurance coverage may compound recovery efforts and negatively impact health equity. The findings highlight the crucial role health insurance plays in access to essential healthcare services, with uninsured individuals facing increased challenges, including those pertaining to the structural determinants of health and engaging with the healthcare system.

Future disaggregated research can explore if insurance coverage remained an issue in subsequent years as the COVID-19 pandemic progressed, as well as if other non-medical challenges persisted for uninsured individuals and how individuals addressed them. Additionally, future research can include measures of social capital (e.g., neighborhood cohesion) to more expansively examine their role in addressing non-medical challenges, as this avenue could provide valuable insights into the interaction between systemic barriers, like lack of insurance, and community-based support systems. Survey research should consider incorporating both an online and a paper component to increase accessibility for a more representative sample, improving the generalizability of results. These future directions can provide comprehensive insights into the healthcare challenges that Vietnamese Americans and other subgroups face within the diverse AANHPI umbrella.

## 5. Conclusions

Limited data exist regarding the impact of the COVID-19 pandemic on health insurance and other non-medical drivers of health within the Vietnamese American community. In this study, lack of health insurance was strongly associated with non-medical difficulties during the COVID-19 pandemic among Vietnamese Americans in Texas. To address these disparities, interventions that improve data disaggregation of Asian American populations, expand language access, and increase support for community-based organizations can help to advance equity for Vietnamese Americans and members of other historically marginalized communities. Policies that address community-specific concerns are critical for strengthening population resilience and health equity, particularly in the context of public health emergencies such as the COVID-19 pandemic. Ultimately, strategies that invest in the long-term infrastructure of historically marginalized communities, such as improving accurate data collection, health literacy, and socioeconomic stability, are effective tactics that can strengthen community resilience and bolster populations against future crises to improve health outcomes.

## Figures and Tables

**Figure 1 ijerph-22-00189-f001:**
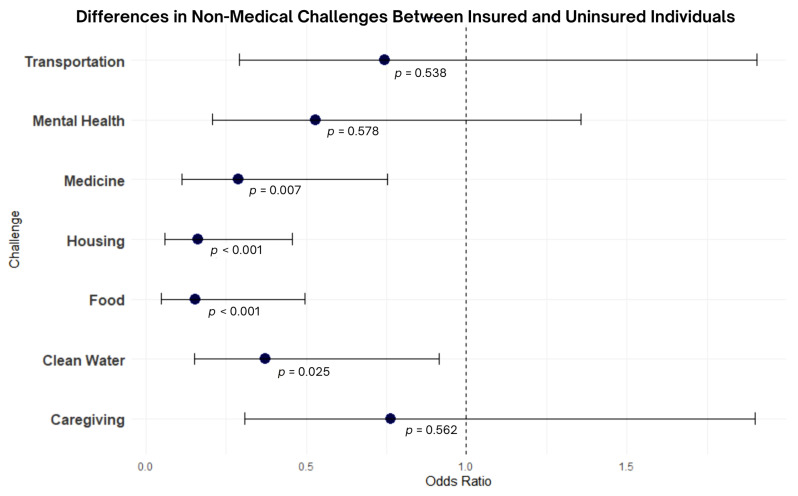
Forest plot comparing differences in non-medical challenges between insured and uninsured individuals.

**Table 1 ijerph-22-00189-t001:** Sociodemographic Characteristics of Survey Respondents.

Characteristic	Participants (*n* = 217)	%
Gender		
Female	126	58.0
Male	83	39.2
Other	6	2.80
Sexual Orientation		
Heterosexual	142	18.4
Homosexual	39	67.0
Other	31	14.6
Age		
18–24 years	23	10.8
25–34 years	40	18.9
35–44 years	36	17.0
45–54 years	36	17.0
55–64 years	35	16.5
65–74 years	34	16.0
75 years and older	8	3.80
Preferred language to use during the survey		
English	160	75.5
Vietnamese	52	24.5
Insurance Status		
Insured	194	87.3
Uninsured	23	10.6
Education		
College degree or higher	104	49.1
Less than a college degree	108	50.9

**Table 2 ijerph-22-00189-t002:** Logistic Regression Examining Differences in Non-Medical Challenges Between Insured and Uninsured Individuals.

	Respondents	Odds Ratio	*p*-Value
Non-Medical Challenges			
Transportation	215	0.746	0.538
Mental Health	211	0.530	0.178
Medicine	211	0.290	0.007
Housing	209	0.164	0.001
Food	213	0.156	0.001
Clean Water	214	0.372	0.025
Caregiving	199	0.765	0.562

## Data Availability

The raw data supporting the conclusions of this article will be made available by the authors on request.

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
