# Peer review of "Examining Health Insurance and Non-Medical Challenges Among Vietnamese Americans in Texas During the COVID-19 Pandemic"

_ijerph, 2025, doi:10.3390/ijerph22020189_

Round 1
Reviewer 1 Report
Comments and Suggestions for Authors
The authors analyse the health insurance coverage and COVID-19-related issues, such as accessing housing, food, and prescriptions, among Vietnamese Americans in Texas.
The work is well written, with a clear objective and flow.
Here are some suggestions I suggest to enhance the manuscript:
1)Please enhance the abstract by referring to the sample rather than the assurance. The purpose is evident in the article, however it is unclear in the abstract.
2)Better describe your sample: How did you choose these at random? Even though they were described in a prior topic, they are still helpful for understanding your analysis in this article. There were 217 out of 224 (97%) replies. This is a very unusual survey outcome. How would you explain this result?
3)I would appreciate some robustness checks: could you please check for collinearity using the Variance Inflation Factor (VIF)? AND, BECAUSE YOU HAVE QUESTIONAIRRES, I WOULD RECOMMEND doing different regressions for subgroups of responders.
4)To improve the findings, consider comparing aggregated data from different ethnic communities. In my perspective, it is not unexpected that a lack of insurance has aggravated COVID-19-related concerns among a certain ethnic group. I find it more fascinating to read statistics that includes other minorities, since this may genuinely highlight the scope of the problem.
5)Furthermore, I would want to use more aggregated data on healthcare access in general to provide a comparison. I'm not sure that this is an issue with insurance and minority groups' access to it. There was a global difficulty with balancing COVID-19-related and non-COVID-related healthcare, which might be an omitted variable in your analysis.
Please refer to Troisi, R., De Simone, S., Vargas, M., and Franco, M. (2022). The second side of the dilemma involves organisational flexibility in balancing Covid-19 and non-Covid-19 health-care services. BMC Health Services Research, 22(1):1096.
6)In your discussion, you mention the ability of Vietnamese Americans to support one another in ensuring health care access and economic resources. I believe it is an interpretive key to emphasise more effectively. According to your findings, those without health insurance have significant challenges with covid. Does this suggest that social inclusion was ineffective in this case?
Reviewer 2 Report
Comments and Suggestions for Authors
The present paper sets out to report health insurance and other “challenges” among Vietnamese Americans during the Covid-19 pandemic. Using a sample drawn from Texas, the authors conclude that a lack of health insurance gave rise to problems accessing care.
At present the paper, while offering some useful information arising from the Covid-19 pandemic, needs considerable work. First, it is not at all clear whether the issues described are as a consequence of the Covid-19 pandemic or simply issues that already exist as a consequence of care delivery arrangements in the United States.
The authors need to develop their rationale in the introduction and highlight more clearly their underlying theory as to why the Covid-19 pandemic has exacerbated the problems of access to care. Indeed, while in the present study Vietnamese Americans have been identified, are other marginalised groups likely to have had the same problems.
Second, the work has been conducted in Texas. Either the title needs to reflect this or the authors provide information to argue that the findings from the present study would apply equally to other areas. While the references provided in the opening would suggest this is the case are there any differences in health care provision in Texas compared to other States.
Given the international readership of the journal, this may be dealt with through a paragraph explaining the care system operating in the States and possible variation.
Third, the sample is relatively small. Given the nature of the population and the overall size suggested by the authors, why? Added to this could the authors explain given the access issues, why did 96% receive at least one vaccination?
Fourth, it is not clear what the authors mean by Covid-19 related challenges. While a list of factors exist is this any different to no-Covid challenges. How exactly did Covid-19 lead to nearly 40% having issues related to clean water for example. What were these issues?
Finally, the use of p>0.5 at the end of the ‘insurance’ paragraph is strange. Surely the authors should be setting a p value to test, the findings from the analyses either accepting or rejecting their hypothesis.
Given the above the authors need to rewrite their paper to help place what could be very useful work in a better context.
Round 2
Reviewer 1 Report
Comments and Suggestions for Authors
It is clear that the authors have taken this revision very seriously, and the improvement is evident. The structure, methodology, and results are now clearer and easier to read, even for a non-expert reader. I suggest that the authors correct the typos in the manuscript and review the references, as new contributions have been added.
Author Response
Dear Reviewer 1,
We sincerely thank you for your considerate review of our revised manuscript. Your suggestions have contributed to improving the clarity of our writing. Below, we responded to your comments.
Reviewer 1: It is clear that the authors have taken this revision very seriously, and the improvement is evident. The structure, methodology, and results are now clearer and easier to read, even for a non-expert reader. I suggest that the authors correct the typos in the manuscript and review the references, as new contributions have been added.
Comment 1: We have corrected typos and updated references.
Reviewer 2 Report
Comments and Suggestions for Authors
This is a much improved resubmission. The authors have addressed previous concerns in a logical and thoughtful manner.
Author Response
Dear Reviewer 2,
We sincerely thank you for your considerate review of our revised manuscript. Your suggestions have contributed to improving the clarity of our writing. Below, we responded to your comments.
Reviewer 2: This is a much improved resubmission. The authors have addressed previous concerns in a logical and thoughtful manner.
Comment 1: We have corrected typos and updated references.